# Dry Two-Step Self-Assembly of Stable Supported Lipid Bilayers on Silicon Substrates

**DOI:** 10.3390/ijms21186819

**Published:** 2020-09-17

**Authors:** Marcelo A. Cisternas, Francisca Palacios-Coddou, Sebastian Molina, Maria Jose Retamal, Nancy Gomez-Vierling, Nicolas Moraga, Hugo Zelada, Marco A. Soto-Arriaza, Tomas P. Corrales, Ulrich G. Volkmann

**Affiliations:** 1Instituto de Fisica, Pontificia Universidad Catolica de Chile, Santiago 7820436, Chile; mncister@uc.cl (M.A.C.); fmpalacios@uc.cl (F.P.-C.); sdmolina@uc.cl (S.M.); ncgomez@uc.cl (N.G.-V.); nhmoraga@uc.cl (N.M.); hizelada@uc.cl (H.Z.); 2Centro de Investigacion en Nanotecnologia y Materiales Avanzados (CIEN-UC), Pontificia Universidad Catolica de Chile, Santiago 7820436, Chile; moretama@uc.cl (M.J.R.); marcosoto@uc.cl (M.A.S.-A.); 3Departamento de Química-Física, Facultad de Quimica y de Farmacia, Pontificia Universidad Catolica de Chile, Santiago 7820436, Chile; 4Departamento de Fisica, Universidad Técnica Federico Santa María, Valparaíso 2390123, Chile; tomas.corrales@usm.cl

**Keywords:** supported lipid bilayers, bio-silica interfaces, self-assembly, phase transitions, artificial membranes, atomic force microscopy, force spectroscopy, high resolution ellipsometry, physical vapor deposition

## Abstract

Artificial membranes are models for biological systems and are important for applications. We introduce a dry two-step self-assembly method consisting of the high-vacuum evaporation of phospholipid molecules over silicon, followed by a subsequent annealing step in air. We evaporate dipalmitoylphosphatidylcholine (DPPC) molecules over bare silicon without the use of polymer cushions or solvents. High-resolution ellipsometry and AFM temperature-dependent measurements are performed in air to detect the characteristic phase transitions of DPPC bilayers. Complementary AFM force-spectroscopy breakthrough events are induced to detect single- and multi-bilayer formation. These combined experimental methods confirm the formation of stable non-hydrated supported lipid bilayers with phase transitions gel to ripple at 311.5 ± 0.9 K, ripple to liquid crystalline at 323.8 ± 2.5 K and liquid crystalline to fluid disordered at 330.4 ± 0.9 K, consistent with such structures reported in wet environments. We find that the AFM tip induces a restructuring or intercalation of the bilayer that is strongly related to the applied tip-force. These dry supported lipid bilayers show long-term stability. These findings are relevant for the development of functional biointerfaces, specifically for fabrication of biosensors and membrane protein platforms. The observed stability is relevant in the context of lifetimes of systems protected by bilayers in dry environments.

## 1. Introduction

Biological membranes are one of the essential components of the cell and, consequently, all living beings. Biological- or plasma- membranes are formed from a series of lipids that act as a selective barrier in cells, bacteria, and many viruses. These lipids self-assemble into bilayer structures, forming phospholipid membranes. In addition of phospholipids, biological membranes are formed by a mixture of other lipids, such as glycolipids and cholesterol. Conventionally, biological membranes are studied under hydrated conditions, but there is also a growing interest in studying their formation and stability in air. In particular, recent attention has focused on understanding the long-term viability of viruses adsorbed on solid surfaces [1]. To study biological membranes and their interaction with solid surfaces in dry conditions, model systems can be created to mimic their properties, i.e., artificial membranes, that have the potential to be used for applications in the field of bionanotechnology [2]. This field combines nanoscale technologies with biological systems in order to create functional devices with diverse applications, such as drug delivery, biosensors, carriers of small molecules and templates for pharmaceutical design [3,4,5,6,7,8,9]. An important goal for bionanotechnology is to create silicon-based chips that can biomimic biological membranes, i.e., bio-silica interfaces [10]. Such a bio-silica interface could be used to study the insertion of proteins within membranes. 

A first step for mimicking a plasma membrane over a solid surface is to form phospholipid bilayers, which are referred to as a supported lipid bilayer (SLB) [11]. One method to form a SLB is by using small vesicles, which are suspended in water or buffer solution and set to interact with a silicon surface. Once the vesicles interact with the hydrophilic silicon surface, they self-assemble into a SLB [12]. There have been many studies reporting different methods to form SLBs from vesicles, such as polymer cushioned lipid bilayers [13,14,15,16], hybrid bilayers supported over alkane-thiol self-assembled monolayers (SAM) [17,18], tethered lipid bilayers [19,20] and freely-suspended bilayers [21,22]. Another method to form SLBs, without starting from a vesicle, is using the Langmuir–Blodgett method [23], where a solid substrate is successively immersed within a compressed layer of lipids, which assemble at a liquid-air interface. Furthermore, a recent solvent-assisted method using isopropanol has been reported to form SLBs, where lipid molecules are dissolved in isopropanol and flowed through a liquid reaction chamber, generating a bilayer over a glass slide [24]. All the aforementioned methods involve the use of solvents to dissolve lipid molecules, transport small vesicles or assemble a transfer monolayer over liquid-air interfaces. The main disadvantage of liquid-based methods is that they are dependent on specific phospholipid and solvent types, concentration and surface tension between the liquid-solid interface. 

Results towards the creation of a solvent-free bio-silica interface was first reported by Retamal et al. (2014) [25], who showed that supported bilayers of dipalmitoylphosphatidylcholine (DPPC) can be evaporated from their vapor phase over a cushion of chitosan (CH), previously evaporated onto the substrate [25]. Using Raman spectroscopy, Retamal et al. showed that the evaporation process did not alter the chemical composition of the DPPC molecules [25]. Furthermore, ellipsometric measurements on these DPPC/CH/SiO_2_/Si samples in air, previously hydrated with a drop of water, showed evidence of gel to ripple, ripple to liquid crystalline and liquid crystalline to fluid disordered phase transitions at 301 K, 315 K and 328 K, respectively. The reported phase transitions are consistent with previous reports in liquid [26,27,28,29,30]. Although promising, some drawbacks of the CH cushion arise due to its dynamics during evaporation. In particular, Retamal et al. showed, in a separate work, that vapor deposited CH exhibits a solid-state dewetting processes on silicon surfaces that occurs during the evaporation at room temperature [31]. Recently, Cisternas et al., reported the vapor deposition of phospholipids on a different biocompatible substrate, titanium (Ti) [32]. After a plasma immersion ion implantation and deposition treatment (PIII&D) a tiny layer of titanium nitride (TiN) was formed and a thin layer of CH was previously vapor-deposited in order to provide a moisturizing matrix for the artificial membrane of DPPC. The CH and, subsequently, the DPPC were vapor-deposited on the plasma-treated TiN substrate. The formation of artificial membranes was confirmed by atomic force microscopy (AFM), measuring the topography at different temperatures and performing force curves [32]. This previous work confirms that the nitrogen PIII&D-treated Ti can be used to support, with enhanced biocompatibility, stable phospholipid artificial membranes (SLBs) that are vapor-deposited in high vacuum. 

In the present work we go one step further and omit the CH moisturizing matrix or cushion. In other words, we focus on evaporating DPPC molecules directly onto a monocrystalline silicon substrate, covered with its native SiO_2_ layer, without further hydration. Our work shows that it is possible to form stable SLBs, which self-assembled without the need of further hydration and which are stable in dry conditions over weeks, months and at least up to nine months. The evaporation of molecules is performed in high vacuum conditions with in situ ellipsometric thickness monitoring. After the evaporation process, the chamber is vented with air and heating ramps at a rate of 2 K/min are performed. Using ellipsometry, we monitor the optical thickness of the samples and observe the characteristic phase transitions of DPPC. In parallel, AFM is used to study the morphological changes of the sample after each heating ramp. Combined AFM and ellipsometric measurements show evidence of the formation of a phospholipid bilayer system that self-assembled in dry conditions. Using AFM, we have performed force curves that show the rupture of several bilayers at increasing loads. Finally, we have observed the restructuring of the bilayer by increasing the AFM force setpoint in imaging mode. The morphology, force-dependent restructuring and rupture forces corroborate the ellipsometric measurements and point to a stable DPPC bilayer in air. 

The finding of this new approach is relevant for the understanding of the interaction of lipid bilayers in contact with surfaces in dry environments, with the aim to develop new kinds of lab-on-chip bionanosensors based on this novel process of SLB formation. This discovery is especially relevant in the context of the viability of organisms covered with lipid bilayer structures. An example of this kind of interaction occurs between viruses and solid surfaces and permits the virus to stay active during long periods of time, which relieves the transfer of virus from visually dry solid surfaces to living hosts. The prolonged stability of SLBs on dry SiO_2_/Si substrates detected in our research can explain for the long-term stability of some viruses deposited or adsorbed on dry surfaces, including the recently appearing SARS-CoV-2 virus.

## 2. Results

In Figure 1 we show an AFM image of DPPC as deposited over silicon and force spectroscopy curves obtained on the different evaporated regions using quantitative imaging (QI^TM^) mode. Figure 1a shows the island-like topography of the sample right after the evaporation process. Figure 1b shows a height distribution of the topography, from where we can identify three clearly separated height levels. From these height distributions we obtain the three main peaks centred at 4.01 ± 0.93 nm (level 1), 8.08 ± 0.94 nm (level 2) and 12.17 ± 0.98 nm (level 3). These step heights are multiples of 4 nm, which is the characteristic height of a DPPC bilayers [26,33]. Figure 1c shows an adhesion map of the same region, while Figure 1d shows an adhesion distribution. From Figure 1d, we can say that the entire surface is covered with a layer that has roughly the same adhesion, i.e., only one peak is seen centred at 9 nN, this is higher than our measured value for the bare silicon substrate, on which we measure an adhesion force of 5 nN (Appendix A).

In order to test the mechanical properties of the membrane in Figure 1, we performed force curves using AFM [34]. Figure 2 shows representative AFM force curves taken using force mapping mode on a 7 µm × 7 µm area. Breakthrough events are seen as sudden drops in the force curves. These breakthrough events are seen at all points, and show 1, 2 or 3 breakthrough events (Figure 2a–c), respectively. Force curves with one breakthrough are attributed to the rupture of the first level, i.e., only one bilayer (Figure 2a). Meanwhile, force curves with two and three breakthrough events are related to ruptures in levels 2 and 3 (Figure 2b,c). The rupture forces show that we have a complete coverage of the substrate with one DPPC bilayer, i.e., we always see at least one rupture event. On top of this complete bilayer we have partial coverage of two and three additional bilayers (see Figure 1a). By averaging all curves with one rupture we obtain an average of 5.3 ± 2.4 nN. Force curves showing two ruptures (level two) show a first rupture at 5.6 ± 1.2 nN followed by a second rupture at 5.1 ± 1.1 nN. Finally, force curves with three rupture events (level 3) rupture first at 5.3 ± 2.0 nN, followed by a second rupture at 5.2 ± 1.8 nN and a third rupture at 5.0 ± 1.7 nN. As seen, all ruptures in dry, non-hydrated SLBs occur around 5nN and are independent of the number of layers. The averages and individual rupture forces are tabulated in Appendix A. To compare the mechanical properties of SLBs immersed in liquid, we applied AFM force curves to phospholipid bilayers fabricated under the same procedure (formed by physical vapour deposition in high vacuum) immersed in pure water in the AFM JPK NanoWizard^TM^ Heating Cooling Module HCM^TM^ accessory. We applied tip forces up to 15 nN and no rupture of the bilayer membrane was observed (see Appendix A).

To confirm the DPPC bilayer formation, complementary ellipsometric measurements were performed in order to observe phase transitions, by applying heating and cooling temperature ramps between room temperature (RT) and 347 K. Figure 3 shows the variations in the polarizer angle (*Δ*P), which represents variations in the polarizability of the bilayer molecules [35]. The ellipsometric data permits, based on the models of Paul Drude [36] and the equivalent model [37] of Dignam and Fedyk [35], an interpretation of the collective behaviour on a molecular scale. Figure 3a corresponds to the first temperature ramp applied to the sample of ~60 Å DPPC on SiO_2_/Si, while Figure 3b,c show the data obtained during the second and third heating ramp, respectively. 

Changes in slope, or inflection points, of the ellipsometric signal indicate changes in the phase of the DPPC bilayers. The three transitions measured in the sample corresponding to Figure 3b,c are from left to right: The gel to ripple at 311 K, ripple to liquid crystalline at 322 K and liquid crystalline to fluid disordered at 330 K, respectively, marked in Figure 3 with vertical dashed lines. Further experiments on four different samples prepared under the same (all curves are shown in Appendix A) conditions give us a gel to ripple transition at 311.5 ± 0.9 K, ripple to liquid crystalline at 323.8 ± 2.5 K and a liquid crystalline to fluid disordered at 330.4 ± 0.9 K (see Appendix A). 

Given that the transition seen at 311 K (gel to ripple) is a smooth continuous transition, we perform complimentary QI^TM^ AFM measurements in this low temperature range. Adhesion maps of 2 × 2 µm were recorded over a total of 3 SLB samples. Temperature ramps were taken using steps of 5 K, starting from room temperature to 328 K. Figure 4a–c show the adhesion maps at 296 K, 308 K and 323 K, for one of the samples. A noticeable change in adhesion is observed between images when compared in the same adhesion range (1–20 nN). By extracting the adhesion distribution at the different temperatures and obtaining the centre and half width of the distribution, we can plot the average adhesion force as a function of temperature, as shown in Figure 4d, for three representative samples. Another adhesion map used to build the graph in the Figure 4d is shown in the Appendix A. As seen in Figure 4d, the adhesion value rises between room temperature and 308–312 K and then drops abruptly. 

For further proof of the formation of dry phospholipid bilayers, we draw on a more detailed study of well-known behaviour of conventionally wet produced SLBs. Attwood et al. [38] and Gosvami [39] described an AFM-tip induced change in morphology of SLBs on mica produced by vesicles in water (DOPC, DPPC) and by the Langmuir–Blodgett (LB) technique (DODAB), respectively. In both cited studies, AFM measurements performed in water show a morphological modification of the scan area due to the interaction of the SLB with the cantilever tip. We analysed the tip-surface interaction as a function of temperature and compared our AFM measurements with temperature-dependent ellipsometric analysis. The latter is a non-invasive method, based on low energy photon-solid interaction, which does not affect the structure of the molecules nor the morphology of the crystalline or fluid ordered phases. In other words, we analysed the topographical channels as a function of temperature to study structural changes in the film. Figure 5 shows AFM topography images before and after temperature ramps, i.e., from room temperature (RT) to 347 K, performed within the ellipsometer (Figure 5a–c) and within the AFM (Figure 5d–f). Figure 5a shows the topography of the evaporated SLBs at RT before the first temperature ramp was performed within the ellipsometer. In Figure 5b,c, we observe the sample at RT after the first and sixth temperature ramp carried out within the ellipsometer, respectively. A reorganization of the initial surface (Figure 5a) into an even more homogeneous bilayer surface is observed (Figure 5b,c). In order to study the reorganization of the sample shown in Figure 5a–c, we perform temperature cycles within AFM to map the surface topography with increasing temperature operating in air. AFM topography images were taken every 5 K, starting from RT up to 343 K (see Appendix A). Figure 5d shows the topography of the evaporated DPPC sample before any temperature ramp. The morphology of Figure 5d is similar to that of Figure 5a. Figure 5e shows the AFM topography of the SLBs after the first temperature ramp was performed within the AFM in the same scanning zone as Figure 5d. Figure 5f shows the AFM topography of the same sample after the first temperature in a contiguous region to the scanned area shown in Figure 5e, i.e., it was not imaged during the temperature cycle. The scanned area during the temperature cycle (Figure 5e) has a roughness parameter of R_q_ = 28 nm, while the contiguous region (Figure 5f) has a roughness of R_q_ = 3.3 nm. 

To quantify the effect of the tip-SLB interaction, we perform AFM imaging in air at RT of the DPPC single- and double bilayers. In Figure 6, we observe the area reduction of a double bilayer and a single bilayer induced by the AFM tip. A total of 10 AFM images were recorded. The first five images were taken at a force setpoint of 1 nN (Figure 6a–c), while the following five were taken at 3 nN (Figure 6d–f), which is below the measured breakthrough force in Figure 2. From the topographical images of Figure 6 (see Appendix A), we observed a clear reduction in the double bilayer area, as well as a restructuring of the single bilayer. 

We measured the area of the DPPC islands seen in Figure 6 by masking the image by height using an image processor (Gwyddion) and obtain the area as a function of scan number, for two different applied tip forces. Figure 7 shows the reduction in area of the DPPC islands as a function of scan number. For scan numbers 1–5, the applied force was 1 nN, and for scan numbers 6–10, 3 nN. The measured area reduction follows a linear behaviour and there is a clear difference in rates between two applied tip forces. For a set point of 1 nN the measured reduction rate was −2.4 × 10^4^ nm^2^/scan and for 3 nN we obtain −6.5 × 10^4^ nm^2^/scan.

We also analysed the depth of the holes created by the AFM tip in the SLBs, as seen in the second bilayer island of Figure 6. The average hole depth is 1.2 ± 0.4 nm, as measured by cross-sections over the hole (see Appendix A). These holes are lower in height than a DPPC monolayer thickness, and do not change with the different setpoints used, i.e., 1 nN and 3 nN. 

## 3. Discussion

The measured AFM rupture forces in Figure 2 in dry conditions (~5 nN) are clearly lower than the reported value in liquid conditions. Rupture forces reported in the literature for a single bilayer of DPPC in physiological conditions are 20 nN, and this rupture force decreases to 13 nN in the absence of Mg^2+^ [40]. Our measurements of the mechanical properties of SLBs immersed in pure water show that the rupture force of the bilayers increased at least three times, compared to dry phospholipid bilayers fabricated under exactly the same procedure (physical vapour deposition in high vacuum). In other words, the hydrated SLBs are stronger than the stable dry SLBs (see Appendix A). On the other hand, the observed stability of these evaporated SLBs, analysed with AFM under water, confirms that the first DPPC bilayer, on top of the SiO_2_/Si substrate, is anchored correctly with its polar head towards the SiO_2_ substrate (head down configuration). In the opposite case, if anchored with the alkane legs towards the substrate (tail down configuration), the bilayer in contact with the substrate would be displaced or removed from the substrate, as we observed during previous studies of alkane layers adsorbed by velocity controlled dip-coating on the same substrate (dotriacontane C_32_H_66_/SiO_2_/Si) [41]. In the case of nonpolar molecules, like alkanes, water intercalates between the hydrophilic SiO_2_ substrate and the alkane molecules and displaces the alkanes, similar to the effect of penetrating oil on water films (WD-40^TM^ or Caramba^TM^ effect). The hydrophilic nature of the silicon substrate, which is defined by the cleaning procedure [42], combined with air humidity, could induce the formation of stable SLBs in air after the first heating cycle. However, this eventual adsorption of water from the laboratory atmosphere is very unlikely, given that the room temperature is above the dew point at the relative humidity of our laboratory (approximately 20%). Furthermore, our extremely sensitive VHRE was not able to detect the adsorption of water molecules underneath, inside or above the bilayer, which would be clearly distinguishable from the DPPC molecules given their different refractive indices and polarizabilities. Additionally, the AFM measurements show no signs of the presence of a significant amount of water.

Our results show that SLBs fabricated by our dry two-step procedure can be used for applications in dry environments as well as for conventional applications under physiological conditions, taking advantage of the extended storage period of our dry and long-time stable SLBs. We believe that water helps the SLB stabilize mechanically, exhibiting higher rupture forces under physiological conditions than in dry environments, as shown in Figure 2.

From the ellipsometry measurements seen in Figure 3, we can observe that after the first temperature cycle the general shape of the heating-cooling curve stabilizes, indicating that the first heating ramp produces an irreversible change in the morphology of the sample, which is maintained throughout the following heating-cooling cycles. In other words, the system is transformed into a stable structure at room temperature in a two-step self-assembly process, i.e., after the first heating ramp. No rearrangements of the molecules are observed in samples stored over at least nine months (see Appendix A). This contrasts to previous observations of alkane monolayers, which are metastable at room temperature, showing mobility that leads to the formation of macro-crystals and depleting areas around them [43]. The phase transitions seen in Figure 3 are characterized by different collective molecule behaviours [44]. The gel phase is an ordered phase formed by slightly inclined phospholipid molecules. In this phase, the ellipsometric signal remains constant. At 311.5 K we see a smooth onset of the gel to ripple transition, characterized by increased molecular dynamics, resulting in an undulated surface and a straightening of the molecules. This straightening of the molecules leads to an increase in the measured polarizability of the molecules, which in our setup is reflected by an increase in the signal. At 323.8 K, the transition from ripple to liquid crystalline (fluid ordered) phase occurs, which implies a collective inclination of the molecules and an increase of gauche defects or kinks in the alkane chains of the phospholipid molecules, dramatically reducing their contribution to the measured polarizability, or ellipsometric signal [37]. At 330.4 K the system starts the transition from liquid crystalline to the fluid disordered phase, where the bilayer structure disappears completely and the surface converts into an almost flat fluid, where the average orientation of the molecules contributes even less to the measured polarizability, or ellipsometric signal. The transition temperatures for ripple to liquid crystalline and liquid crystalline to fluid disordered suggests the existence of a structural first-order phase transition [45,46]. 

AFM adhesion measurements in Figure 4 exhibit two regimes, one where the adhesion increases between room temperature and ~308–313 K and a regime where the adhesion drops above 313 K. From the literature we know that the temperature range of the gel-ripple transition corresponds to 306–310 K [26]. We believe that the increase of adhesion seen in Figure 4 between room temperature up to 308–312 K, is related to the appearance of ripples in the membrane. These ripples have a reported wavelength of 14 nm [47], which is comparable to the AFM tip radius, which means the tip could fit partially within the groves, generating an increase in contact area and, therefore, in adhesion, as compared to a flat and unwrinkled membrane. The three curves in Figure 4 show a maximum range between 308 and 312 K, after which the adhesion shows a sharp decrease related to a phase transition. This temperature range falls within the reported values for the ripple to liquid crystalline phase transition [26]. We believe that this sharp drop is related to the ripple to liquid crystalline transition given that the membrane smoothens, i.e., loses its ripples, generating a decrease in the contact area with the tip.

Figure 5 AFM topography images lead us to believe that the AFM tip is interacting with the SLB in air, affecting its morphology and roughness, during the heating cycle. Restructuring of an SLB on mica due to tip interactions has been reported on DODAB, DOPC and DPPC on mica in liquid [38,39]. It is important to emphasize that in the AFM images seen in Figure 5a–d and in Figure 5f, that the first layer, corresponding to a height of 0 nm, contains a complete phospholipid bilayer, which was confirmed by the observation of a single rupture event in the force curves. Meanwhile in Figure 5e there is no DPPC bilayer in the first height layer, since it was removed and piled up due to the interaction between the AFM tip and the sample during the temperature cycle. This was confirmed by force curves which do not show a rupture event. 

The holes seen in the SLBs in Figure 6 were measured by taking cross-sections that have a depth of ~1.2 nm, which is independent of the loading force. We believe these holes are due to a local interdigitation or intercalation of the alkane legs or gauche defects/kinks in the lipid molecules hydrocarbon chains. From these measurements we can see that the SLB is a dynamic system that interacts with the AFM tip in air, even though the applied forces are below any possible rupture of the membrane (<5 nN). 

Figure 7 shows the reduction rate dependence on the external tip force. We emphasize that these results are time-independent. The observed reduction only depends on the number of scans and on the intensity, i.e., force setpoint 1 nN or 3 nN, of the “hammer blows” executed by the AFM tip. During these measurements in QI^TM^ mode, only vertical forces act on the surface. In other words, there are no lateral forces involved, except on the borders of the bilayer islands. This means we are not dragging material across the scanned area, so we can rule out this effect on the reduction dynamics seen in Figure 7.

## 4. Materials and Methods

### 4.1. Sample Preparation

As a substrate we used monocrystalline silicon wafers (100) covered with their native oxide layer (15−25 Å), obtained from Virginia Semiconductor Inc. (Fredericksburg, VA, USA). Substrates were cleaned in a mixture of H_2_SO_4_ (sulphuric acid 95−97%) and H_2_O_2_ (hydrogen peroxide 30%), both acquired from Merck KGaA (Darmstadt, Germany), in a 7:3 ratio at 363 K during 30 min (piranha chemical bath) [42], rinsed twice and stored in ultrapure water (Merck). Before use, the substrates were dried by applying a jet of ultrapure nitrogen gas (99.995% purity) purchased from Linde (Santiago, Chile). Cleaning efficiency was confirmed using a home-built very high resolution ellipsometer (VHRE), able to detect even traces of organic contamination [48,49].

### 4.2. Physical Vapour Deposition with In Situ Ellipsometric Monitoring

Dry DPPC in powder form, purchased from Avanti Polar Lipids Inc. (Alabaster, AL, USA), was allocated in a Knudsen cell inside the high vacuum evaporation chamber, without any further treatment (out of the box). The evaporation process was started in high vacuum (∼10^–6^°Torr). During evaporation, at approximately 443 K inside the Knudsen cell, the pressure in the chamber increased up to ∼10^–4^ Torr. Substrate temperature was kept at room temperature (∼296 K), and the deposition rates were approximately 3.2 × 10^−2^ Å/s.

The optical thickness of each thin film was monitored in situ with our home-built VHRE, which has an incidence angle of 60.5° with respect to the sample’s normal direction. During the phospholipid deposition process in high vacuum, the variation of the absolute ellipsometric polarizer angle P of the sample was measured using an electronic feedback loop connected to a Faraday rotator. This polarizer angle (P) increases during the deposition process with respect to an initial clean angle P_0_ ~47°, with a fixed analyser angle A_0_ ~20°, as described in [31]. The layer thickness is then calculated using the Drude model for single layers by relating the polarizer (P) and analyser (A) angles to the ellipsometric parameters *Δ* = 2P + 90° and ψ = A [37]. After deposition, the high vacuum chamber was vented with air at atmospheric pressure and the sample was analysed as a function of the sample temperature with VHRE and/or taken out of the chamber for further analysis with AFM.

Temperature ramps from room temperature (RT) to 347 K were performed within the ellipsometer setup at a heating rate of 2 K/min.

### 4.3. Atomic Force Microscopy

We used an AFM (JPK NanoWizard 3, Bruker Nano GmbH, Berlin, Germany) to obtain topographic information and mechanical properties of the prepared films. AFM cantilevers, model PPP-CONTSCAuD, were purchased from Nano and More (Watsonville, CA, USA). The dimensions of our cantilevers were: thickness, 1 μm; length, 225 μm; and width, 48 μm. The resonance frequency was ∼40 kHz, and the force constant was 0.6–1.5 N/m. AFM measurements were performed using QI^TM^ mode, yielding images of 256 × 256 pixels. Each QI^TM^ image was taken at a vertical speed of 45 µm/s, z-length of 112 nm and a rate of 2.4 ms per pixel. In this mode each pixel corresponds to a complete force distance curve. The images were analysed using Gwyddion (Czech Metrology Institute, Brno, Czech Republic) (v.2.54) and JPK Data Processing software (Bruker Nano GmbH, Berlin, Germany) (v.spm-6.1.49). Using Gwyddion, we treated all AFM images with the following processes: Level data by mean plane subtraction and/or level data to make facets point upward; align rows using various methods (e.g., median method and/or trimmed mean method); correct horizontal scars; shift the minimum data value to zero; stretch the colour range to part of the data (explicitly set the fixed colour range and/or automatic colour range with tails cut off). Force mapping mode (also known as force volume) was used to collect an array of force curves containing mechanical properties of the samples, which can also lead to spatial reconstruction of topographic maps [50]. The JPK QI^TM^ mode was used to obtain topographical images and adhesion maps. In addition, force curves were measured using force mapping mode in order to achieve a higher precision, due to an increased control of the parameters such as vertical speed and force. Force mapping mode images were taken on 7 × 7 µm scan areas, obtaining 64 force curves over the region. 

## 5. Conclusions 

In this work, we have shown a dry two-step method for SLB formation, consisting in thickness-controlled phospholipid deposition in high vacuum onto bare SiO_2_/Si substrates, followed by a subsequent heating cycle performed in air. The SLBs are stable in air without the application of water or buffer solution to the sample. These vapour-deposited SLBs exhibit phase transitions, which we measure using VHRE in air, that are consistent with literature values of SLBs in liquid. The observed phase transitions lead us to believe that our SLBs are biologically functional. AFM measurements carried out in air exhibit rupture forces which are roughly 5 nN. These rupture forces are lower than values we measured in liquid (>15 nN) for samples prepared under the same conditions. This confirms that water has a stabilizing effect on the mechanical properties of the membranes. Our control measurements in water corroborate that the first monolayer of the vapour-deposited stable DPPC bilayer assembles with the polar heads towards the substrate. We have observed that our SLBs tend to restructure in air when interacting with the AFM tip, even though the applied forces are lower than the rupture force. Our evaporated membranes show long-term stability and no restructuring after storage in air for at least nine months. This extreme stability of the studied SLB structure makes this system interesting for technical applications in the field of functional biointerfaces, e.g., for the fabrication of biosensors and membrane protein platforms, including cleanroom-compatible fabrication technology. Our SLBs could also help us gain insight into the lifetime of viral structures protected by a surrounding phospholipid bilayer adsorbed on static solid surfaces or on inhalable particulate material (PM), which contributes to the spread of the SARS-CoV-2 virus.

## Figures and Tables

**Figure 1 ijms-21-06819-f001:**
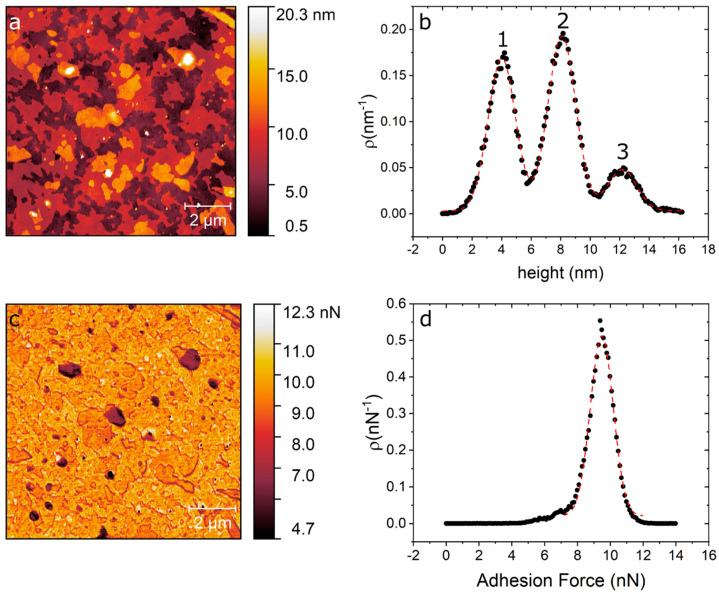
(**a**) AFM topography image of DPPC/SiO_2_/Si taken directly after evaporation. (**b**) Height distribution of the topography map showing the three distinct levels at 4 nm, 8 nm and 12 nm. (**c**) Adhesion map of the same region. (**d**) Adhesion distribution centred at 9 nN.

**Figure 2 ijms-21-06819-f002:**
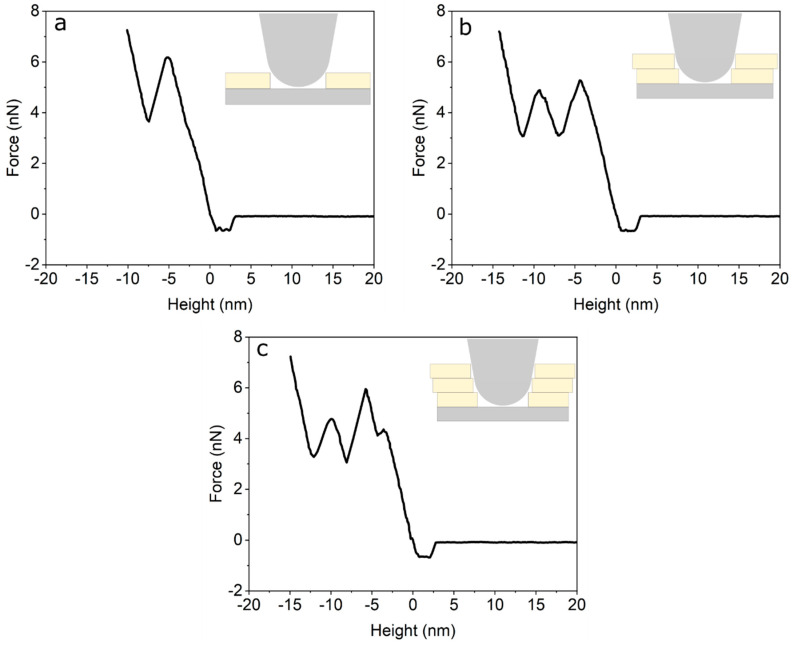
(**a**) Force curve on the first bilayer (level 1). (**b**) Force curve on the second bilayer (level 2). (**c**) Force curve on the third bilayer (level 3).

**Figure 3 ijms-21-06819-f003:**
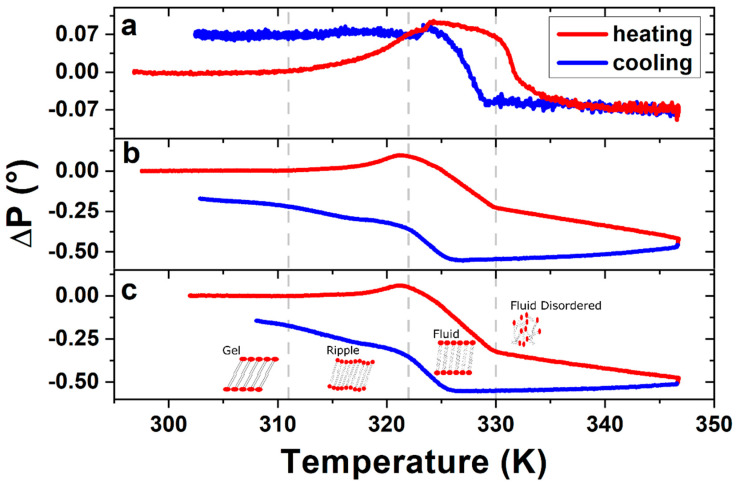
Ellipsometric changes in polarizer angle ΔP of DPPC bilayer, measured during (**a**) the first, (**b**) the second and (**c**) the third heating–cooling cycle. Vertical dashed lines indicate the phase transition temperatures as described in the text. The ΔP signal of the following heating–cooling cycles show the same characteristics of figures (**b**) and (**c**), leading to the conclusion that the structure formed after the first heating–cooling cycle remains stable.

**Figure 4 ijms-21-06819-f004:**
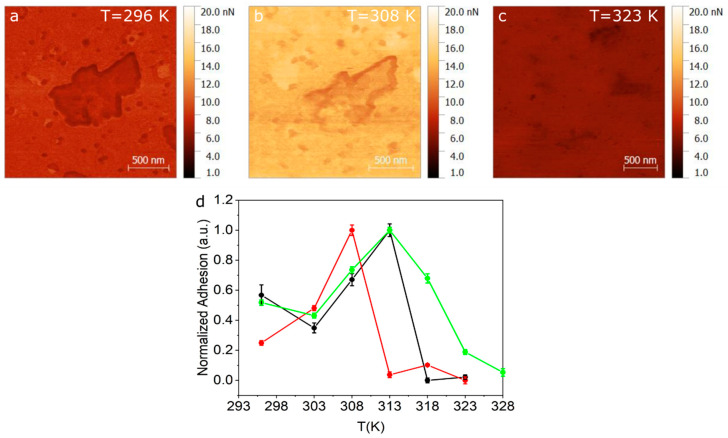
Adhesion maps obtained using QI^TM^ mode for (**a**) 296 K (**b**) 308 K and (**c**) 323 K. (**d**) Normalized adhesion taken for three different samples as a function of temperature.

**Figure 5 ijms-21-06819-f005:**
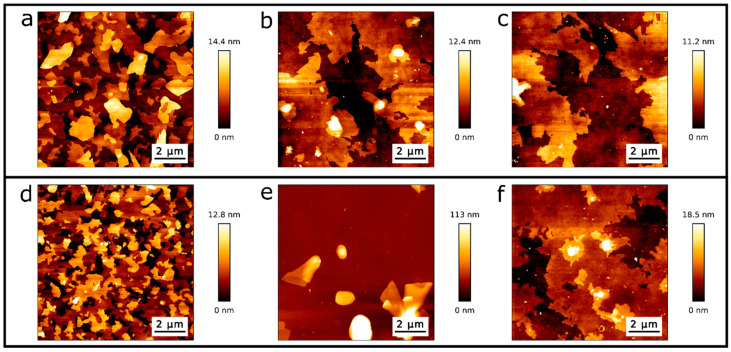
Topographic AFM images from DPPC/SiO_2_/Si samples at RT: (**a**) directly after evaporation and before the first temperature cycle, (**b**) after the first and (**c**) sixth temperature cycle performed within the ellipsometer. (**d**) Topography image directly after evaporation and before temperature cycling within the AFM, (**e**) after the first temperature cycle while scanning with AFM, and (**f**) after the first temperature cycle in a contiguous region to the scanned area.

**Figure 6 ijms-21-06819-f006:**
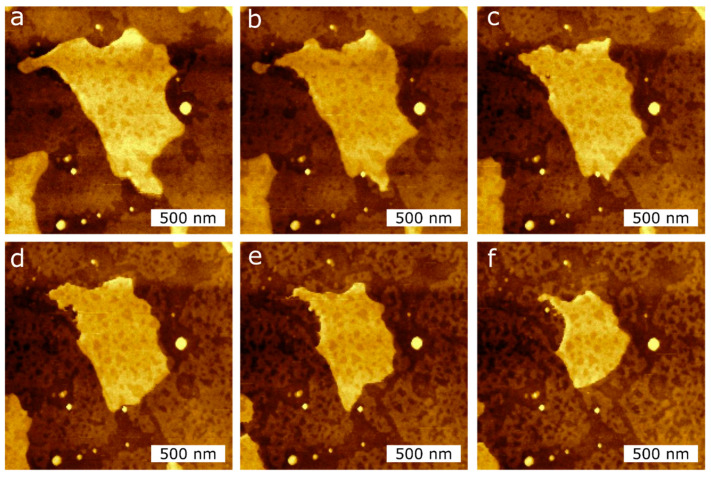
AFM topography images of the DPPC bilayer taken at RT in air during consecutive measurements in the same scan region (2 µm × 2 µm) to induce the reduction of the island-like DPPC bilayer surface area, applying two different tip forces, (**a**–**c**): 1 nN; (**d**–**f**): 3 nN.

**Figure 7 ijms-21-06819-f007:**
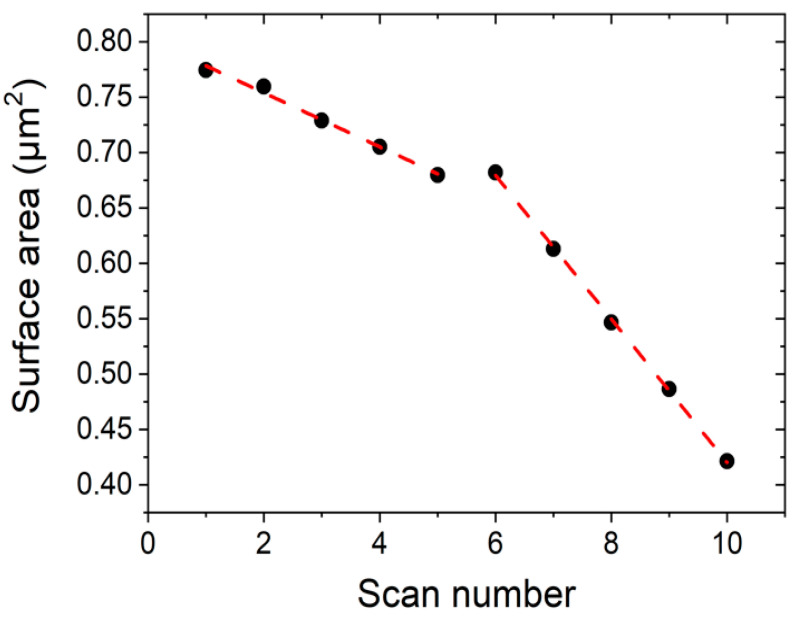
Surface area reduction during consecutive scans at RT using two different tip forces applied in the same scan regions. Scans 1–5 are taken at 1 nN, while scans 6–10 are taken at 3 nN.

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
