# Peer review of "Dry Two-Step Self-Assembly of Stable Supported Lipid Bilayers on Silicon Substrates"

_ijms, 2020, doi:10.3390/ijms21186819_

Round 1

Reviewer 1 Report

Marcelo A. Cisternas et al. in the manuscript "Dry two-step self-assembly of stable supported lipid bilayers on silicon substrates" proved that it is possible to form supported lipid bilayers in a dry solvent-free process by physical vapor deposition without further hydration. The authors claimed that created by them bilayers are stable in dry conditions for up to a year.

They research is particularly important in the context of the viability of organisms covered with lipid bilayer structures. The long-term stability of stable supported lipid bilayers on dry SiO2/Si substrates can lead to an explanation of the long-term stability of some virus deposited or adsorbed on dry surfaces.

I have only one more serious comment for the manuscript test. The conclusions are not properly written. They cannot be a summary of the studies presented earlier in the paper. They are also far too long.

And one more small comment. The caption for Figure 7 should be Figure 7, not Figure 1.

I strongly recommend this paper for publication in International Journal of Molecular Sciences because the authors present proper methodology and interesting results, and significant conclusions are drawn in the manuscript. This review should be of interest and utility to workers in a number of branches of the field.

Author Response

Point 1: I have only one more serious comment for the manuscript test. The conclusions are not properly written. They cannot be a summary of the studies presented earlier in the paper. They are also far too long.

Response 1: Line 411 – 430: We re-wrote and shortened the conclusion section. We moved some of the conclusions to the discussion section (line 288 – 296), taking care not to duplicate information. We reduced the length of the conclusion section by more than half (from 620 to 281 words).

The conclusion now reads:

In this work, we have shown a dry two-step method for SLB formation, consisting in thickness-controlled phospholipid deposition in high vacuum onto bare SiO2/Si substrates, followed by a subsequent heating cycle performed in air. The SLBs are stable in air without the application of water or buffer solution to the sample. These vapour deposited SLBs exhibit phase transitions, which we measure using VHRE in air, that are consistent with literature values of SLBs in liquid. The observed phase transitions lead us to believe that our SLBs are biologically functional. AFM measurements carried out in air exhibit rupture forces which are roughly 5nN. These rupture forces are lower than values we measured in liquid (>15nN) for samples prepared under the same conditions. This confirms that water has a stabilizing effect on the mechanical properties of the membranes. Our control measurements in water corroborate that the first monolayer of the vapour deposited stable DPPC bilayer assembles with the polar heads towards the substrate. We have observed that our SLBs tend to restructure in air when interacting with the AFM tip, even though the applied forces are lower than the rupture force. Our evaporated membranes show long-term stability and no restructuring after storage in air for at least nine months. This extreme stability of the studied SLB structure makes this system interesting for technical applications in the field of functional biointerfaces, e.g. for fabrication of biosensors and membrane protein platforms, including cleanroom-compatible fabrication technology. Our SLBs could also help us gain insight into the lifetime of viral structures protected by a surrounding phospholipid bilayer adsorbed on static solid surfaces or on inhalable particulate material (PM), which contributes to the spread of the SARS-CoV-2 virus.

Point 2: And one more small comment. The caption for Figure 7 should be Figure 7, not Figure 1.

Response 2: We renamed the wrong figure number 1 into figure number 7.

Reviewer 2 Report

This manuscript contains interesting results that I recommend for publication at IJMS. The presentation in the form of this submission would benefit from some important revision effort.

In the main manuscript please refer:

  • In the Materials and Methods section, the Gwyddion version and the JPK Data Processing version you are using.
  • Also, given that JPK was acquired by Bruker the correct nomination now would be: JPK NanoWizard 3, Bruker Nano GmbH, Berlin, Germany.
  • Although it is mention the characteristics of the cantilever, I did not find the name itself. Which cantilever have you used from NanoAndMore? This information is important for the readers of the manuscript.
  • The Quantitative Imaging mode has specific acquisition parameters. Please refer in the description of the AFM, in the materials and methods section, the Z-cantilever velocity, the Z-length and the respective acquisition times. These information is easily obtained in the data information file of the images.
  • What kind of AFM image processing have you applied? What levelling? Please describe it. This is important information that has to be included.
  • There is a lack of consistency in naming the abbreviations along the manuscript. For example, the first time it appears chitosan is in line 74 and the full name and abbreviation appears on line 94. Despite there is a list of abbreviations at the end of the article, please make it uniform for all the manuscript such that all abbreviations are firstly referenced in full name in the manuscript, like for example in line 120, QI – Quantitative Imaging (QI).
  • What ellipsometer instrument (brand) have you used? Is it home-built? What were the angles chosen for the data acquisition?
  • Do you have AFM evidences of stability over one year? The manuscript would benefit for a comparison AFM image for example to be included for example in supporting information.
  • As a last suggestion, in Figure 2, it would help the readers of the manuscript if you include a schematic diagram of the interaction of the tip with the different levels of the SLB’s.

In the Supporting Information:

Figure S 11, the picture at 343K. This picture is difficult to interpret. What structures are represented in there? In particular at around 8 nm?

Author Response

Comments and Suggestions for Authors

This manuscript contains interesting results that I recommend for publication at IJMS. The presentation in the form of this submission would benefit from some important revision effort.

In the main manuscript please refer:

Point 1: In the Materials and Methods section, the Gwyddion version and the JPK Data Processing version you are using.

Response 1: We add in line 398 and 399 the Gwyddion version (2.54) and the JPK Data Processing version (spm-6.1.49) we are using.

Point 2: Also, given that JPK was acquired by Bruker the correct nomination now would be: JPK NanoWizard 3, Bruker Nano GmbH, Berlin, Germany.

Response 2: We add in line 391 and in abbreviations the updated information about the AFM Brand.

Point 3: Although it is mention the characteristics of the cantilever, I did not find the name itself. Which cantilever have you used from NanoAndMore? This information is important for the readers of the manuscript.

Response 3: We add in line 392 - 395 the information about the cantilevers we used for this research, model PPP-CONTSCAuD, with a small range of spring constants between 0.6 and 1.5 N/m, from NanoAndMore USA Corporation, 21 Brennan St, Suite 10, Watsonville, CA 95076.

Point 4: The Quantitative Imaging mode has specific acquisition parameters. Please refer in the description of the AFM, in the materials and methods section, the Z-cantilever velocity, the Z-length and the respective acquisition times. These information is easily obtained in the data information file of the images.

Response 4: In line 396 to 398 we add the specific information about the acquisition parameters. Each QITM image was taken at a vertical speed of 45 µm/s, z-length of 112 nm and a rate of 2.4 ms per pixel. In this mode each pixel corresponds to a complete force distance curve.

Point 5: What kind of AFM image processing have you applied? What levelling? Please describe it. This is important information that has to be included.

Response 5: In line 399 - 403 we indicate all processing steps that were applied, using Gwyddion, to AFM images: Level data by mean plane subtraction and / or level data to make facets point upward; align rows using various methods (e.g. median method and / or trimmed mean method); correct horizontal scars; shift minimum data value to zero; stretch colour range to part of data (explicitly set fixed colour range and / or automatic colour range with tails cut off).

Point 6: There is a lack of consistency in naming the abbreviations along the manuscript. For example, the first time it appears chitosan is in line 74 and the full name and abbreviation appears on line 94. Despite there is a list of abbreviations at the end of the article, please make it uniform for all the manuscript such that all abbreviations are firstly referenced in full name in the manuscript, like for example in line 120, QI – Quantitative Imaging (QI).

Response 6: We revised and corrected the naming of all abbreviations along the manuscript to make sure that all abbreviations are firstly referenced in full name in the main text of the manuscript.

Point 7: What ellipsometer instrument (brand) have you used? Is it home-built? What were the angles chosen for the data acquisition?

Response 7: In line 378 - 387  we describe and give references to our home-built Very High Resolution Ellipsometer in PCSA (Polarizer – Compensator – Sample – Analyser) null configuration, with a fixed angle of incidence (60,5º) and an electronic feedback loop including a Faraday Rotator, which permits to maintain fixed the motor-driven mechanical prism rotators and to increase the resolution due to reading of the corresponding Faraday currents or voltages, proportional to change of polarizer angle. The device is described in Retamal, et al., Biomacromolecules 2016, 17, 3, 1142–1149, doi.org/10.1021/acs.biomac.5b01750 and references cited within this paper (Cisternas, E. A.; Corrales, T. P.; del Campo, V.; Soza, P. A.; Volkmann, U. G.; Bai, M.; Taub, H.; Hansen, F. Y. Crystalline-to- Plastic Phase Transitions in Molecularly Thin N-Dotriacontane Films Adsorbed on Solid Surfaces. J. Chem. Phys. 2009, 131 (11), 114705). Measurements start at the Polarizer and Analyser angle corresponding to clean Si wafer substrate covered with its native oxide layer (typically ~47º and ~20º for Polarizer and Analyzer, resp.). Film growth during deposition and phase transitions during heating cooling cycles are tracked by measuring the variations in the Faraday Current.

Point 8: Do you have AFM evidences of stability over one year? The manuscript would benefit for a comparison AFM image for example to be included for example in supporting information.

Response 8: In supporting information, figure S14, we have added an AFM image and histogram of the height distribution of a sample taken two hours after evaporation and an AFM image and histogram of the same sample taken 9 months later.

Due to the Covid-19 pandemic and the related shut down of our Campus in Santiago de Chile, we have not had access to our labs since March 16, 2020, and therefore have not been able to revise our data nor perform new AFM measurements on these samples which now have more than 18 months.

Point 9: As a last suggestion, in Figure 2, it would help the readers of the manuscript if you include a schematic diagram of the interaction of the tip with the different levels of the SLB’s.

Response 9: We add to Figure 2 a schematic diagram of the interaction of the tip with each of the three different levels of the SLBs.

Point 10: In the Supporting Information:

Figure S 11, the picture at 343K. This picture is difficult to interpret. What structures are represented in there? In particular at around 8 nm?

Response 10: At 343 K the SLB is in the fluid disordered phase and the AFM cantilever accumulates material at the right side of the image.